# Tuberculosis Preventive Treatment in People Living with HIV in Uganda: Facilitators and Barriers for Initiation and Completion

**DOI:** 10.3390/tropicalmed10110303

**Published:** 2025-10-27

**Authors:** Ritah Mande, Pruthu Thekkur, Denis Mudoola, Joseph Nsonga, John Paul Dongo, Simon Muchuro, Stavia Turyahabwe, Henry Luzze, Proscovia Namuwenge, Selma Dar Berger, Deus Lukoye, Macarthur Charles, Odile Ferroussier-Davis, Riitta A. Dlodlo

**Affiliations:** 1International Union Against Tuberculosis and Lung Disease, Plot 2, Lourdel Road, Nakasero Hill, Kampala P.O. Box 16094, Uganda; 2International Union Against Tuberculosis and Lung Disease, 2 Rue Jean Lantier, 75001 Paris, France; 3The National Tuberculosis and Leprosy Program, Ministry of Health, Plot 6, Lourdel Road, Nakasero, Kampala P.O. Box 7749, Uganda; 4AIDS Control Program, Ministry of Health, Plot 6, Lourdel Road, Nakasero, Kampala P.O. Box 7272, Uganda; 5Centers for Disease Control and Prevention, Kampala P.O. Box 7007, Uganda; 6Centers for Disease Control and Prevention, Atlanta, GA 30333, USA

**Keywords:** tuberculosis preventive treatment, PLHIV, 4S screening, TPT initiation, TPT adherence, TPT completion, operational research

## Abstract

Tuberculosis preventive treatment (TPT) is a mainstay for reducing the tuberculosis (TB) burden among people living with human immunodeficiency virus (PLHIV). Context-specific challenges hinder TPT uptake and completion among PLHIV. During 2022–2024, a mixed-methods design was used to evaluate the TPT cascade and explore its facilitators and barriers among PLHIV availing care from 12 PEPFAR-supported health facilities in Uganda. The quantitative component included analysis of routine programmatic data, and the qualitative component included focus group discussions and in-depth interviews with healthcare workers and PLHIV. A total of 1349 PLHIV were enrolled in the evaluation. Among PLHIV newly initiated on ART (≤3 months), 74% started TPT, and 98% of them completed it. In PLHIV already on ART, 87% had initiated TPT (76% before and 11% during this evaluation), with a treatment completion rate of 98%. The facilitators for TPT implementation included access to shorter TPT regimens, integration of services, and adherence counseling. Barriers included knowledge gaps, pill burden, TPT drug stock-outs, and documentation inconsistencies. The TPT completion rate was higher than the national target (90%), but the TPT initiation remains low. Improved access to shorter regimens, adherence counseling, better documentation, and service integration can sustain the completion rate and improve the initiation rate in Uganda and possibly elsewhere.

## 1. Introduction

Although tuberculosis (TB) is largely preventable and curable, it remains the leading infectious cause of death among people living with HIV (PLHIV). In 2023, an estimated 161,000 PLHIV died from TB worldwide, accounting for one-third of HIV-related deaths [1]. TB preventive treatment (TPT) has long been a recommended component of collaborative HIV-TB activities [2,3].

In 2018, at the United Nations High-Level Meeting (UNHLM) on TB, member states set the ambitious target of providing TPT to 30 million high-risk individuals, including PLHIV, and the U.S. President’s Emergency Plan for AIDS Relief (PEPFAR) committed to providing TPT to all PLHIV on antiretroviral therapy (ART) in PEPFAR-supported programs by September 2021 [4,5]. Despite ongoing efforts, TPT implementation remains suboptimal. By 2022, only 15.5 million, or 52% of the UNHLM target, had received TPT [6]. Even among PLHIV initiated on TPT, the TPT completion rates remain poor, especially in low- and middle-income countries (LMIC) [7,8,9,10,11]. Poor completion rates have been attributed to longer TPT duration and drug-associated toxicity [8,12,13]. Recent advances have added shorter treatment regimen options (one month of daily isoniazid and rifapentine [1HP] and three months of weekly isoniazid and rifapentine [3HP]) with better completion rates than the six-month isoniazid regimen (6H) and comparable effectiveness [14,15,16].

Uganda is a high TB-HIV burden country. In 2023, 96,000 persons with TB were notified, a third of whom were HIV-positive [1]. Uganda has adopted the TPT guidelines issued by the World Health Organization (WHO) and has implemented several initiatives to scale up TPT among PLHIV, including the introduction of shorter regimens [17]. However, the TPT coverage among PLHIV is just 6.9% [1], as reported in 2024, largely because of unresolved implementation challenges.

In an effort to address these challenges, an evaluation was conducted to generate evidence to inform strategies to enhance the capacity of healthcare workers (HCWs) to improve TPT provision at selected health facilities. The objectives of this study included describing the TB and TPT care cascades among PLHIV and identifying the facilitators and barriers to TPT initiation and completion as perceived by both HCWs and PLHIV. By understanding these dynamics, the study aimed to support the development of targeted strategies that can effectively promote TPT uptake and ultimately improve health outcomes for this vulnerable population.

## 2. Materials and Methods

### 2.1. Study Design

This was a concurrent mixed-methods study [18]. The quantitative component was a retrospective review of routine programmatic data. The qualitative component used a descriptive design with focus group discussions (FGD) and in-depth interviews with PLHIV and healthcare providers to explore facilitators and barriers to TPT initiation and completion.

### 2.2. Study Setting

The health system in Uganda consists of both public and private sectors, with each contributing approximately 50% of the overall service delivery. The National TB and Leprosy Program (NTLP) in the Ministry of Health (MoH)’s Department of National Disease Control manages the provision of TB care through public- and private-sector facilities. According to the national guidelines [17], PLHIV should be screened for TB at every visit to ART clinics, using the 4-symptom (4S) complex (cough of any duration, fever, weight loss, and night sweats). PLHIV who report any of the four symptoms are evaluated for TB using the Xpert MTB/RIF assay, sputum smear microscopy, urine TB-LAM test, and/or chest X-ray, and if diagnosed with TB, are started on anti-TB treatment [17].

PLHIV are eligible to receive TPT if they do not have TB disease and have no contraindications, such as jaundice or peripheral neuropathy. In PLHIV initiating a dolutegravir-based ART regimen, TPT is delayed for three months. PLHIV diagnosed with TB initiate TPT after completing anti-TB treatment. All PLHIV receive adherence counseling at TPT initiation. For 6H, adults/adolescents receive isoniazid 300 mg and pyridoxine (vitamin B6) 25 mg daily for 6 months; children ≥12 months receive isoniazid 100 mg and pyridoxine 25 mg daily for 6 months. For 3HP, people weighing ≥10 kg receive rifapentine 300 mg and isoniazid 300 mg once a week for three months. In children <12 months of age, TPT is recommended only if the child is in contact with someone with TB disease [17].

TPT medications are dispensed together with ART. While on TPT, PLHIV are screened for TB symptoms at every visit to the health facility. If TB is diagnosed, TPT is stopped, and anti-TB treatment is started. Patients are monitored for adverse events (AE) after TPT initiation based on their (or their caregiver’s) description of symptoms suspected to be associated with TPT drugs. Drug discontinuation is based on the severity of AE.

TB screening and TPT provision are documented by the HCWs in multiple records and registers. The 4S screening results (including TB status) are documented in the ART register and ART card. TPT initiation and completion dates are recorded in the TPT register, ART card, and ART register. AEs due to a TPT regimen are recorded on the pharmacovigilance form and the ART card and reported to the National Drug Authority. PLHIV with TB disease are recorded in the TB register.

### 2.3. Study Location

The study was conducted in 12 health facilities supported by the Centers for Disease Control and Prevention (CDC) through PEPFAR in four high TB-HIV burden regions (Masaka, Mubende, Hoima, and Kampala). Facility selection used purposive sampling to reflect a mix of rural/urban settings, private/public health facilities, district hospitals, and lower-level facilities (Figure 1).

### 2.4. Capacity Building

The International Union Against Tuberculosis and Lung Disease, in collaboration with the MoH, conducted training, support supervision, and on-site mentorship of HCWs at the selected health facilities on programmatic management of TB infection, including recording and reporting of TB screening and TPT provision. This was conducted over 18 months at quarterly intervals.

### 2.5. Study Population

#### 2.5.1. Quantitative

Adults and children already on ART (≥3 months) and newly initiated (<3 months) were eligible for the study. Facility-level sample size targets for each category (and separately for children [<15 years of age] and adolescents and adults [≥15 years] within each category) were set based on existing programmatic data on TB screening, TPT uptake, and completion among PLHIV. For each group, enrollment of eligible PLHIV was carried out as they visited the health facilities during routine appointments until the target sample size at each facility was reached.

#### 2.5.2. Qualitative

Purposive sampling was used to select healthcare workers (n = 32) involved with TPT implementation for ≥6 months. Criterion-based purposive sampling was used to select adult PLHIV (aged ≥ 18) (n = 12) at different stages of TPT (completed TPT, initiated but did not complete TPT, and did not receive TPT) care provision.

### 2.6. Variables, Data Sources, and Data Collection

#### 2.6.1. Quantitative

Data abstraction was conducted over 18 months at quarterly intervals using a pre-tested standardized data collection tool. The first data abstraction was conducted in November 2022, and the last one in May 2024. Data on TPT indicators included the percentage of PLHIV who did not have TB disease at ART initiation, percentage of PLHIV without TB disease who underwent 4S screening at ART initiation, percentage of PLHIV with presumptive TB who underwent TB evaluation, percentage eligible for TPT, percentage initiated on TPT, and the percentage who completed TPT. Data sources included the ART patient cards, presumptive TB registers, TB laboratory registers, TB registers, TPT registers, ART registers, MDR-TB registers, and pharmacovigilance forms. Where data were missing or needed confirmation, the electronic medical records (EMR) were used.

#### 2.6.2. Qualitative

The FGDs and in-depth interviews were conducted between October 2022 and June 2024 by three researchers (two males and one female) trained in qualitative research methods and fluent in the local languages (Luganda and Runyoro) and English. The interview and FGD guides were prepared in English, Luganda and Runyoro. The interviews that were conducted in the local languages were translated into English during note taking since the interviewers were multi-lingual. Three FGDs involving participants from all twelve facilities were conducted, with each FGD comprising six to eight participants. The in-depth interviews were conducted in private at the health facilities. All FGDs and in-depth interviews were were audio-recorded. On average, the FGDs lasted for 60 min, and the in-depth interviews lasted for 30 min. The number of participants interviewed was guided by the saturation of findings.

### 2.7. Data Entry and Analysis

#### 2.7.1. Quantitative

The abstracted data were entered into an EpiData Entry Client (v4.7.0, Odense, Denmark). The data were analyzed using STATA statistical software (version 16.0, StataCorp LLC, College Station, TX, USA). TB and TPT care cascades were characterized according to the following indicators: the number of PLHIV enrolled, the number screened for TB and found to have symptoms of TB disease, the number with sputum specimens collected for laboratory testing, the number with bacteriologically confirmed and clinically diagnosed TB, the number started on anti-TB treatment, anti-TB treatment outcomes, the number eligible for TPT, and the number who started and completed TPT. The median and interquartile range were used to describe the duration between TPT initiation and ART initiation, among those who were initiated on TPT.

Unadjusted (using univariate binomial regression) and adjusted (using cluster-adjusted [health facility] multivariable binomial regression) relative risks (RR) were calculated to measure the strength of association between demographic factors (age group, sex, region, and duration of ART) and TPT initiation among PLHIV already on ART and new on ART. To evaluate the precision of these estimates, 95% confidence intervals (CI) were computed for the relative risks.

#### 2.7.2. Qualitative

The FGD and in-depth interview data were transcribed and uploaded into the Atlas.ti software (version 9.1.7.0, L-9B8-D20) for storage, organization, coding, and analysis. An inductive coding approach was applied, allowing themes and categories to emerge directly from the data. This facilitated a grounded understanding of the facilitators and barriers to TPT initiation and completion as perceived by participants. The data were coded by author RM, and the codes were reviewed by PT and OF. Codes and categorization reflect consensus among the three researchers. Analysis aimed to identify patient-level, provider-level, and system-level facilitators and barriers for TPT initiation and completion as perceived by HCW and PLHIV.

### 2.8. Ethics

The protocol for this evaluation was approved by the Uganda Joint Clinical Research Centre’s Research Ethics Committee and registered with the National Council for Science and Technology (registration number: HS2165ES on 27 September 2023). This protocol was reviewed by the CDC, deemed not research, and conducted consistent with applicable federal law and CDC policy (45 C.F.R. part 46, 21 C.F.R. part 56; 42 U.S.C. Sect. 241(d); 5 U.S.C. Sect. 552a; 44 U.S.C. Sect. 3501 et seq.). Informed consent was obtained from all participants in FGDs and interviews.

## 3. Results

### 3.1. Quantitative Findings

In total, 1349 PLHIV were enrolled, including 603 newly initiated on ART and 746 already on ART. The characteristics of enrolled PLHIV are provided in Table 1.

#### 3.1.1. PLHIV New on ART

Among 603 PLHIV newly initiated on ART, 58 (10%) were children (aged < 15 years) and 345 (57%) were female. Thirteen (2%) were on anti-TB treatment at ART initiation (Figure 2). Of those not on anti-TB treatment, 585 (99%) received 4S screening at ART initiation, of which 55 (9%) had presumptive TB. Of these, 31 (56%) were diagnosed with TB (21 bacteriologically confirmed, 10 clinically diagnosed). All 31 initiated anti-TB treatment.

Among 44 PLHIV initiated on anti-TB treatment before or at ART initiation, 42 (95%) successfully completed it; 10/42 (24%) initiated TPT (four on 6H, six on 3HP) by the evaluation censor date (May 2024). The median (IQR) time to TPT initiation after completing anti-TB treatment was 24 (1–69) days. All 10 who initiated TPT after completing their anti-TB treatment completed TPT by the evaluation censor date.

Of the 554 PLHIV without TB disease during ART initiation, 409 (74%) initiated TPT by the censor date. Among them, 195 (48%) received 6H, 194 (47%) received 3HP, and 12 (3%) received 1HP. The median (IQR) duration from ART initiation to TPT initiation was 105 (71–152) days. The TPT initiation rate varied from 53% to 89% across the health facilities. The Hoima region had significantly lower TPT initiation rates (adjusted relative risk [RR]: 0.67, 95% CI: 0.63–0.71) compared to the Kampala region (Table 2). The 6H, 3HP, and 1HP completion rates were 97%, 98%, and 100%, respectively.

#### 3.1.2. PLHIV Already on ART

Among 746 PLHIV already on ART, the median (IQR) age was 26 (9–39) years, 430 (58%) were females, and 271 (36%) had been on ART for ≥6 years when enrolled into this evaluation (Table 1). Seven (1%) had been on TB treatment at the time of ART initiation (Figure 3). Of the remaining 739 (99%), 706 (96%) were screened for TB symptoms, of which 50 (7%) screened positive and were further evaluated. Twenty-five (50%) were diagnosed with TB; all were initiated on TB treatment. Of the 32 who had TB at ART initiation, 31 (97%) had successfully completed anti-TB treatment, of which 24 (77%) initiated TPT by the evaluation censor date. Of those, 23 (96%) received 6H. The median (IQR) time to TPT initiation after completing TB treatment was 33 (14–53) days. Of 24 PLHIV initiated on TPT, 23 completed it, and 1 had TB by the censor date.

Among the 681 PLHIV already on ART without TB disease at the time of ART registration, 519 (76%) had already initiated TPT by the time of enrolment into the evaluation, and another 75 (11%) initiated TPT during the evaluation. The TPT initiation rate before enrolment was significantly higher among those aged 30 to 44 years (aRR: 1.19, 95% CI: 1.02–1.39) compared to those aged 15 to 29 years, and among those with longer duration in ART care compared to those in care for 4 to 6 months. The Mubende region (aRR: 0.65, 95% CI: 0.43–0.97) had a significantly lower TPT initiation rate before enrolment compared to the Kampala region. The TPT initiation rate before enrolment ranged from 33% to 92% across facilities (Table 3).

Of the 594 PLHIV already on ART that started TPT, 542 (91%) received 6H, 38 (6%) received 3HP, and 14 (2%) had no record on regimen at the time of evaluation. The median (IQR) duration between ART initiation and TPT initiation was 391 (100–1275) days. Among the 220 PLHIV who were initiated on TPT and had been on ART for less than four years before enrollment into the evaluation, the median (IQR) duration between ART initiation and TPT initiation was 121 (57–231) days. The completion rate for 6H, 3HP, and those whose regimen was not recorded were 99%, 100%, and 64%, respectively.

No adverse events were reported among PLHIV, likely due to documentation gaps: minor events are recorded in patient charts but not always entered into the register.

### 3.2. Qualitative Findings

#### 3.2.1. TPT Initiation—Facilitators

Eleven codes related to facilitators for TPT initiation were deduced from the FGD and interview transcripts (Appendix A). The eleven codes were grouped into nine categories under the subthemes of patient-, provider-, and health system-related facilitators (Figure 4).

Patient-level facilitators: Patients highlighted the importance of positive feedback from peers and of TPT’s perceived benefits as motivating factors for accepting to initiate TPT. Patients also mentioned being influenced by HCWs whom they trust for their expertise and knowledge (Appendix A).

Provider-level facilitators: HCWs explained that continuous medical education (CME) sessions on TPT have increased their awareness of TPT’s benefits and their confidence in initiating TPT for eligible patients. Given the benefits of TPT, the HCWs feel it is important to give high priority to TB prevention to reduce TB disease. This knowledge has translated into proactivity in flagging all persons who are eligible in the register and enrolling them on TPT. Also, HCWs reported taking advantage of various community activities to initiate TPT while conducting home visits, especially for those PLHIV unable to visit the health facilities.

System-level facilitators: From HCWs’ viewpoint, the availability of shorter TPT regimens has improved TPT acceptance among PLHIV and has facilitated timely TPT initiation. The integration of TPT in HIV diagnostic and care provision has increased access to services, and the availability of clear SOPs for TPT has reduced the burden of care among PLHIV.

#### 3.2.2. TPT Initiation—Barriers

Eleven codes related to barriers for TPT initiation were deduced from the FGD and interview transcripts (Figure 4).

Patient-level barriers: Patients’ lack of awareness of TPT and its benefits was perceived to hinder TPT initiation. Some experienced financial and time constraints and missed appointments. Some HCWs felt that the pill burden and patients’ fear of side effects were key reasons for the latter’s non-acceptance of TPT.

Provider-level barriers: HCWs felt their knowledge gaps about TPT contributed to low TPT initiation. HCWs also mentioned hesitancy to initiate some eligible PLHIV on TPT due to the pill burden, especially among PLHIV on treatment for other comorbidities. Also, heavy workloads due to understaffing hindered them from timely eligibility assessment and initiation.

System-level barriers: Non-availability of gastric lavage and investigations, such as liver function tests, for assessing eligibility among PLHIV, as well as TPT stock-outs, were reported as barriers for initiating TPT. HCW also mentioned that their clinic had been participating in clinical trials on the efficacy of newer TPT regimens that had rigid inclusion criteria for TPT initiation, which resulted in non-initiation of TPT in persons among whom it was difficult to rule out TB.

#### 3.2.3. TPT Completion—Facilitators

Eight codes related to facilitators for TPT adherence and completion were deduced from the transcripts (Figure 5) (Appendix A).

Patient-level facilitators: Patients’ beliefs that TPT offers benefits, including the reduced risk of TB, enhanced quality of life, and improved health, motivated PLHIV to complete TPT. Additionally, the lack of side effects further facilitated adherence to TPT.

Provider-level facilitators: According to HCWs, home visits to patients who miss appointments ensured continuity of treatment as the patients were able to receive their drug refills at their homes. Additionally, offering adherence counseling to patients motivated them to adhere to and complete their TPT courses. Finally, HCWs’ adherence to SOPs on AE management, which encouraged identifying and managing AEs due to TPT, also motivated patients to complete their TPT.

System-level facilitators: Availability of shorter TPT regimens reduced the pill burden, thus motivating PLHIV to adhere to and complete TPT. Linking patients to treatment supporters enabled them to receive comprehensive support, including reminders to take the medicines, hence improving adherence. Finally, Differentiated Service Delivery (DSD) models have aligned ART and TPT resupply appointments, consequently reducing the number of required visits to the facility.

#### 3.2.4. TPT Completion—Barriers

Ten codes related to barriers for TPT adherence and completion were deduced from the transcripts (Figure 5).

Patient-level barriers: HCWs identified poverty and the inability of families to afford food as barriers to TPT adherence and completion. Another barrier was PLHIV’s prioritization of other medications, such as ART, over TPT. Attrition from care due to undocumented self-transfers, especially among migratory populations, also hindered TPT completion. HCWs also highlighted stigma around TPT due to prevailing myths (that TPT is only for PLHIV), pill burden, and reluctance among caregivers of children living with HIV (CLHIV) as barriers to completion.

Provider-level barriers: HCWs identified a shortage of staff as a challenge that resulted in gaps and inconsistencies in recordings in the registers and EMR. They considered adherence monitoring difficult to verify because it depends on the information provided by the PLHIVs taking TPT.

System-level barriers: Lack of TPT-adherence tracking tools left HCWs to assume that the PLHIV took the TPT as advised. HCWs also pointed to the lack of short TPT regimens for children, among whom adherence was suboptimal. Lastly, resource constraints limited the facilities’ ability to provide community drug refills.

## 4. Discussion

The findings from the study add to the global evidence on facilitators and barriers to TPT initiation, adherence, and completion. There are four key findings. First, it is noteworthy that more than 95% of PLHIV are screened for TB at the time of ART initiation. Second, only three-fourths of eligible PLHIV were initiated on TPT due to patient-, provider-, and system-level barriers. Third, 6H was predominantly offered as the TPT regimen for PLHIV due to the limited rollout of 3HP in some facilities. Finally, the TPT completion rate was high, largely because of adherence support by HCWs, supportive supervision by The Union and MOH staff, and availability of shorter regimen in the selected health facilities.

The study has several strengths. First, it addresses the global and national research priority on TB prevention among PLHIV. Second, the mixed-methods design adopted in the study enabled a comprehensive assessment of TPT implementation and the underlying reasons for gaps in implementation. Third, findings from the routinely collected program data reflect the real-world realities of implementation. Finally, the study adhered to Strengthening the Reporting of Observational studies in Epidemiology (STROBE) and Consolidated criteria for Reporting Qualitative research (COREQ) guidelines for conducting and reporting the study findings [19,20].

The study has some limitations. First, the inconsistencies in documentation processes across health facilities prevented the extraction of key data elements, and despite consulting multiple data sources, clinical records, especially those related to TPT initiation, adverse events, adherence, and TPT completion, were still incomplete or missing. This limitation may affect our ability to draw conclusions from some of the findings. For example, no documentation of TPT initiation in any of the patients registered was considered as a non-receipt of the TPT, which might have led to an underestimation of TPT coverage due to poor documentation. Secondly, inclusion of PLHIV who visited the health facilities during the recruitment period might have led to selection bias, with inclusion of PLHIV, those who might be relatively adherent to the health facility visits and might have had a higher chance of initiating and completing the TPT. Thus, the selection bias might have contributed to the overestimation of TPT initiation and completion. Thirdly, the TPT completion status was recorded in the registers as reported by the PLHIV without objective assessment using methods like pill count. Patients might have exaggerated their adherence and completion of TPT to meet the perceived expectations of the HCWs. This social desirability bias might have led to overestimation of the TPT completion rate. Fourthly, there might have been a loss of information during the qualitative interviews due to participants’ inability to recall all the relevant issues related to TPT initiation and completion. Lastly, the study was conducted in a limited number of purposively selected facilities, which may affect its representativeness and generalizability. The findings may not fully capture variations in different settings, potentially limiting broader applicability to diverse populations and healthcare contexts.

Despite these limitations, the study findings have some important programmatic implications. First, the study revealed that approximately 98% of the PLHIV were screened for symptoms suggestive of TB (4S screening), indicating high adherence to the screening guidelines by the HCWs. This has been facilitated by the integration of TB screening into routine care at the different entry points, such as ART centers, and training HCWs to conduct TB screening. This high 4S screening has facilitated the initiation and uptake of TPT among PLHIV. Similar findings of the benefit of integration of TB services in ART care have been reported from Uganda and other countries [8,12,21,22].

Second, TPT was initiated only in three-fourths of the PLHIV in care. Some of the PLHIV might not have been initiated on TPT due to contraindications, and the TPT initiation rate might have been underestimated. However, the quantification of PLHIV not eligible for TPT due to contraindications was not possible with the current recording system. Similar TPT initiation rates have been reported in studies from Africa and Asia [8,12,23]. In Uganda, the current national guidelines recommend TPT for PLHIV only after three months of ART initiation to reduce pill burden, ensure adherence to ART drugs during the initial phase of treatment, and avoid overlapping toxicities [17]. To initiate TPT after three months, HCWs have to diligently track PLHIV, screen them again for TB, and initiate TPT if eligible. Given their heavy workload, HCWs may fail to track PLHIV using paper-based records and initiate TPT. Also, many PLHIV opt for proxy ART refill (ART refill on someone’s behalf) as early as three months after initiating ART. The facilities may fail to initiate TPT in these PLHIV, as it is necessary to screen for TPT eligibility in-person and provide counseling before initiating TPT. Similar challenges for TPT initiation have been reported in India [8]. These challenges can be overcome by instituting PLHIV-tracking systems to alert HCWs of PLHIVs’ eligibility for TPT assessment after three months of ART initiation [24].

Third, 6H was the most common TPT regimen offered to PLHIV, and it can adversely impact TPT initiation and completion rates as reported elsewhere. The pill burden and long duration with 6H are major reasons for resistance from PLHIV to initiate and complete TPT [8,13,23]. Also, TPT is delayed for three months after initiating ART among PLHIV in Uganda, risking them of TB disease within this period. Improving access to shorter regimens, which can enhance acceptance of TPT among PLHIV, and considering early initiation of TPT at the time of ART initiation [14,15,16], are avenues for action. The increased use of 3HP noted among PLHIV new on ART (47%), compared to those already on ART (6%), included in the evaluation, is a promising development.

Fourth, it is important to note the high TPT completion among PLHIV. The completion rates for both the shorter regimen and the 6H regimen were excellent. Overall, the TPT completion rate surpassed the national target (90%). This is consistent with findings from a study conducted in Uganda in 2022, which showed a national TPT completion rate of 92% [25]. This may be attributed to HCW training in adherence counseling, supportive supervision during the evaluation, and availability of shorter regimens, notably 3HP, in the selected health facilities. The NTLP of Uganda should consider scaling up these good practices for improving the TPT completion among PLHIV in the country. LMICs with poor TPT completion rates among PLHIV can also adopt these practices, especially by improving access to shorter regimens. However, this would require additional resource allocation for TPT implementation among PLHIV.

Fifth, no adverse events among PLHIV on TPT were reported, primarily due to documentation gaps in the evaluation facilities. Minor adverse events that do not require pausing or interrupting TPT are often not recorded in TPT registers, even though they have a designated space for such entries. Unsatisfactory quality of data in registers can hinder effective monitoring and treatment optimization. Weaknesses in data recording may be attributable to health staff not coping with multiple paper-based registers in the routine programmatic setting. Thus, there is an urgent need for simplifying the recording and reporting system of TPT implementation among PLHIV and considering expansion of digital medical records and registers at subnational-level health facilities, and strengthening point-of-care data capture [24].

## 5. Conclusions

This comprehensive evaluation of TB diagnosis and TPT cascade in selected health facilities of Uganda showed gaps in TPT initiation but high treatment completion rates. These findings point to clear programmatic priorities. Strengthening patient education and peer support can address demand-side barriers to TPT initiation. Enhancing health information systems—through digitized records and tracking tools—may help providers systematically identify and follow up eligible PLHIV, addressing delays in initiation. Ensuring consistent drug supply and expanding shorter regimen availability are critical for sustaining high completion rates. Future efforts should prioritize addressing the identified barriers at all levels by introducing innovative solutions, such as digitalizing medical records with decision support systems, mobile-based adherence tracking tools, and community-based delivery of TPT to improve TPT uptake and completion among PLHIV.

## Figures and Tables

**Figure 1 tropicalmed-10-00303-f001:**
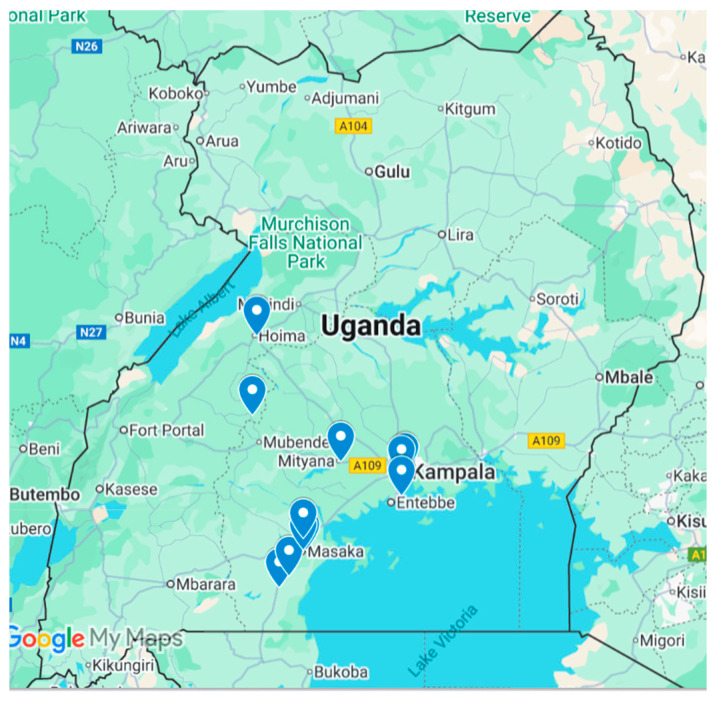
Health facilities included in the study.

**Figure 2 tropicalmed-10-00303-f002:**
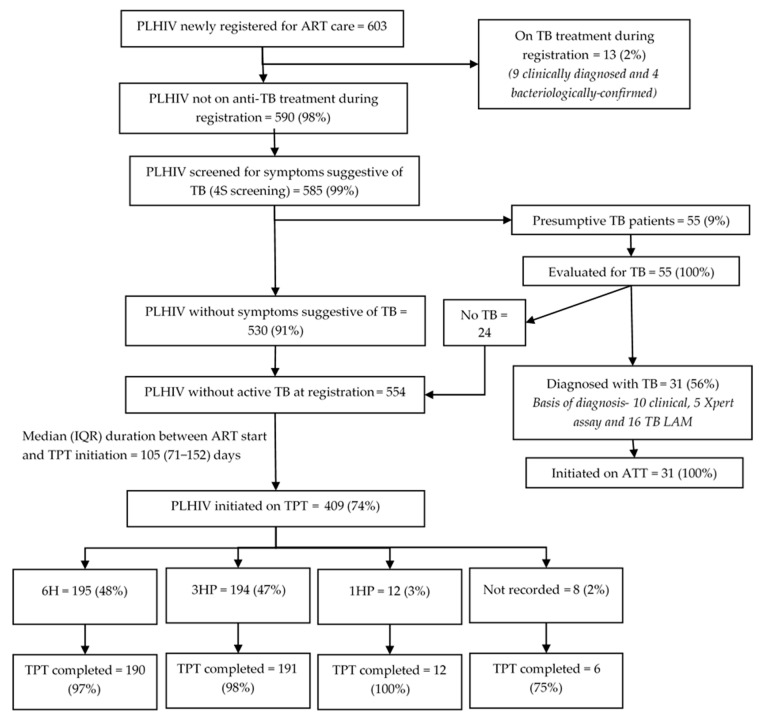
TB screening and TPT cascade among PLHIV newly initiated on ART (<3 months of ART) in 12 selected health facilities in Uganda, 2022–2024. Abbreviations: PLHIV—people living with HIV; TB—tuberculosis; ATT—anti-TB treatment; TB LAM—tuberculosis lipoarabinomannan assay; TPT—TB preventive treatment; 6H—six months of isoniazid; 3HP—three months of isoniazid and rifapentine; 1HP—one month of isoniazid and rifapentine.

**Figure 3 tropicalmed-10-00303-f003:**
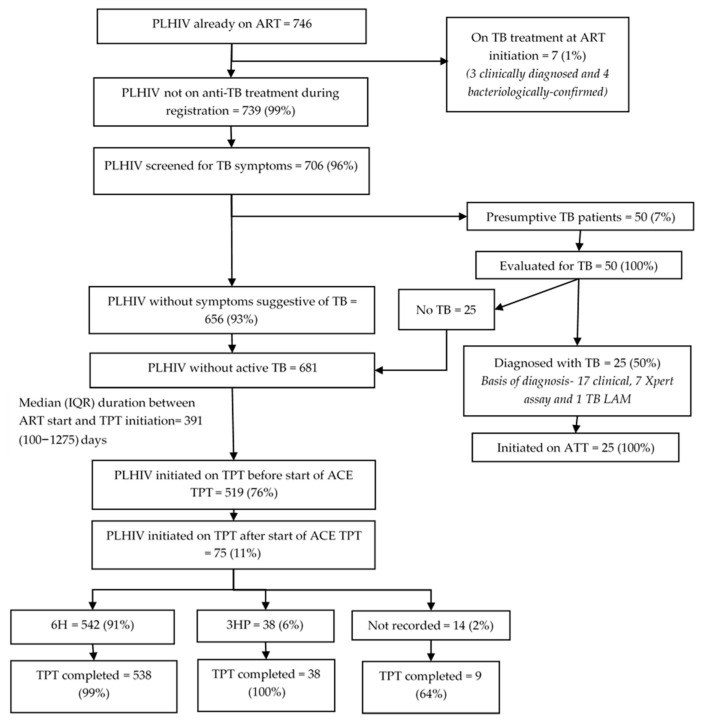
TB screening and TPT cascade among PLHIV already on ART (on ART for ≥3 months at enrolment into the evaluation) in 12 selected health facilities in Uganda, 2022–2024. Abbreviations: PLHIV—people living with HIV; TB—tuberculosis; ATT—anti-TB treatment; TB LAM—tuberculosis lipoarabinomannan assay; TPT—TB preventive treatment; 6H—six months of isoniazid; 3HP—three months of isoniazid and rifapentine; 1HP—one month of isoniazid and rifapentine.

**Figure 4 tropicalmed-10-00303-f004:**
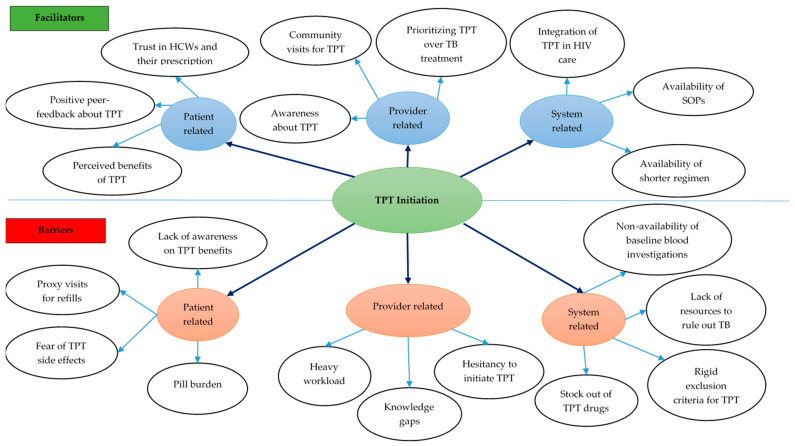
Facilitators and barriers for TPT initiation among PLHIV in 12 selected health facilities in Uganda, 2022–2024. Abbreviations: TB—tuberculosis; SOP—standard operating procedure; HCW—healthcare workers; TPT—TB preventive treatment.

**Figure 5 tropicalmed-10-00303-f005:**
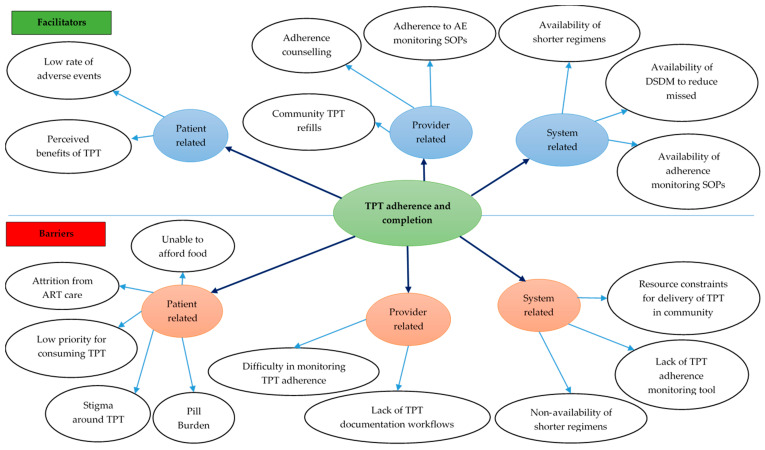
Facilitators and barriers for TPT completion among PLHIV in 12 selected health facilities in Uganda, 2022–2024. Abbreviations: AE—adverse event; ART—antiretroviral therapy; DSDM—differentiated service delivery model; SOP—standard operating procedure; TPT—TB preventive treatment.

**Table 1 tropicalmed-10-00303-t001:** Demographic characteristics of PLHIV newly initiated on ART (<3 months on ART) and already on ART (initiated on ART ≥ 3 months before study enrolment) in 12 selected health facilities in Uganda, 2022–2024.

	Newly Initiated on ART		Already on ART	
Characteristics	n	(%) *	n	(%) *
Total	603	(100)	746	(100)
Age in years				
Median (IQR)	30 (23–38)		26 (9–39)	
<5	34	(5.6)	85	(11.4)
5–14	24	(4.0)	209	(28.0)
15–29	226	(37.5)	120	(16.1)
30–44	246	(40.8)	221	(29.6)
45–59	61	(10.1)	93	(12.5)
≥60	12	(2.0)	18	(2.4)
Sex				
Male	258	(42.8)	316	(42.4)
Female	345	(57.2)	430	(57.6)
Region				
Kampala	321	(53.2)	439	(58.9)
Masaka	171	(28.4)	220	(29.5)
Mubende	34	(5.6)	39	(5.2)
Hoima	77	(12.8)	48	(6.4)
Time on ART at enrollment into the evaluation				
Median (IQR) in years			3.3 (1.2–6.5)	
0–3 months	603	(100.0)		
4–6 months			56	(7.5)
7–12 months			102	(13.7)
1–3 years			194	(26.0)
4–5 years			123	(16.5)
6–10 years			202	(27.1)
>10 years			69	(9.3)

* Column percentage; ART—antiretroviral therapy.

**Table 2 tropicalmed-10-00303-t002:** Demographic factors associated with initiation of TPT among PLHIV newly initiated on ART (<3 months on ART) without TB disease in 12 selected health facilities in Uganda, 2022–2024.

Characteristics	Total	Initiated on TPT		uRR	(95% CI)	aRR	95% CI
		n	(%) *				
Total	554	409	(73.8)				
Age (in years)							
≤14 (Children)	50	36	(72.0)	1		1	
15–29	209	147	(70.3)	0.98	(0.80–1.18)	1.02	(0.84–1.43)
30–44	230	180	(78.3)	1.08	(0.90–1.31)	1.11	(0.88–1.40)
45–59	56	40	(71.4)	0.99	(0.78–1.26)	1.02	(0.85–1.22)
≥60	9	6	(66.7)	0.93	(0.57–1.52)	0.96	(0.68–1.38)
Sex							
Male	235	177	(75.3)	1.04	(0.94–1.14)	1.02	(0.94–1.11)
Female	319	232	(72.7)	1		1	
Region							
Kampala	299	236	(78.9)	1		1	
Masaka	154	112	(72.7)	0.92	(0.82–1.03)	0.93	(0.82–1.05)
Mubende	29	23	(79.3)	1.00	(0.82–1.22)	1.00	(0.94–1.07)
Hoima	72	38	(52.8)	**0.67**	**(0.53–0.84)**	**0.67**	**(0.63–0.71)**

* Row percentage; TPT—TB preventive treatment; uRR—unadjusted relative risk; aRR—adjusted relative risk; ART—antiretroviral therapy.

**Table 3 tropicalmed-10-00303-t003:** Demographic factors associated with TPT initiation before study enrollment among PLHIV already on ART without TB disease in 12 selected health facilities in Uganda, 2022–2024.

Characteristics	Total	Initiated on TPT		uRR	(95% CI)	aRR	(95% CI)
		n	(%) *				
Total	681	519	(76.2)				
Age (in years)							
≤14 (Children)	278	219	(78.8)	**1.21**	**(1.04–1.41)**	1.14	(0.97–1.35)
15–29	108	70	(64.8)	1		1	
30–44	196	157	(80.1)	**1.23**	**(1.06–1.44)**	**1.19**	**(1.02–1.39)**
45–59	86	64	(74.4)	1.15	(0.95–1.38)	1.10	(0.88–1.39)
≥60	13	9	(69.2)	1.07	(0.72–1.57)	0.99	(0.71–1.38)
Sex							
Male	287	300	(76.1)	1.00	(0.92–1.09)	1.01	(0.96–1.07)
Female	394	219	(76.3)	1		1	
Region							
Kampala	399	306	(76.7)	1		1	
Masaka	197	166	(84.3)	**1.09**	**(1.01–1.19)**	1.09	(0.99–1.20)
Mubende	38	17	(44.7)	**0.58**	**(0.41–0.83)**	**0.65**	**(0.43–0.97)**
Hoima	47	30	(63.8)	0.83	(0.67–1.04)	0.90	(0.81–1.02)
Time on ART at enrollment into the evaluation							
4–6 months	44	15	(34.1)	1		1	
7–12 months	87	63	(72.1)	**2.11**	**(1.38–3.27)**	**1.98**	**(1.38–2.84)**
1–3 years	184	148	(80.4)	**2.36**	**(1.56–3.58)**	**2.16**	**(1.32–3.53)**
4–5 years	120	101	(84.2)	**2.47**	**(1.63–3.75)**	**2.23**	**(1.43–3.49)**
6–10 years	191	149	(78.0)	**2.29**	**(1.51–3.48)**	**2.08**	**(1.36–3.19)**
>10 years	55	43	(78.8)	**2.29**	**(1.49–3.54)**	**2.12**	**(1.27–3.56)**

* Row percentage; TPT—TB preventive treatment; uRR—unadjusted relative risk; aRR—adjusted relative risk; ART—antiretroviral therapy.

## Data Availability

The data presented in this study are available on request from the corresponding author as the data are not publicly available due to privacy restrictions.

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
