# Peer review of "Tuberculosis Preventive Treatment in People Living with HIV in Uganda: Facilitators and Barriers for Initiation and Completion"

_tropicalmed, 2025, doi:10.3390/tropicalmed10110303_

Round 1
Reviewer 1 Report
Comments and Suggestions for Authors
Setences in first person should be avoided. "We analysed... we observed... " must be changed to "It was analysed..."
The space-time distribution must be declared in the work by graphs and not in tables, so I suggest changing some of the items in table 1 to location graphs.
Figure 1, 2 and 3 are difficult to read. Please enlarge or change the font color (white is not recommended).
The work would be much more enriching and enlightening if most of the results were not presented in tables. In a work with a large demographic and social data component, graphs and maps are more advisable.
The conclusions are general and not focused on the reality presented. Please complete them and consider community education and the One Health concept.
Author Response
Comment 1: Sentences in first person should be avoided. "We analysed... we observed... " must be changed to "It was analysed..."
Response 1: Thank you for the comment. Throughout the manuscript we have made edits to avoid first person reporting.
Comment 2: The space-time distribution must be declared in the work by graphs and not in tables, so I suggest changing some of the items in table 1 to location graphs.
Response 2: Thank you for the comment. As we already have three tables and four figures in the manuscript, we had to bring some of the data into tables instead of creating multiple graphs. However, we appreciate the comment of the reviewer on including the location graph (map) for depicting the study sites. We have added this in the revised manuscript.
Comment 3: Figure 1, 2 and 3 are difficult to read. Please enlarge or change the font color (white is not recommended).
Response 3: Thank you for the comment. As suggested by the learnt reviewer, we have changed the color of the Figures 1, 2 and 3. Changes are highlighted in the revised manuscript.
Comment 4: The work would be much more enriching and enlightening if most of the results were not presented in tables. In a work with a large demographic and social data component, graphs and maps are more advisable.
Response 4: Thank you for the comment. We agree on the reviewer’s comment. However, as we already have three tables and four figures in the manuscript, we had to bring some of the data into tables instead of creating multiple graphs. Based on the reviewer’s comment, we have included additional map in the manuscript.
Comment 5: The conclusions are general and not focused on the reality presented. Please complete them and consider community education and the One Health concept.
Response 5: Thank you for the comment. We have made necessary changes in the conclusion and have highlighted the need for community education.

Reviewer 2 Report
Comments and Suggestions for Authors
This is a well-structured and relevant paper addressing a high-priority public health issue in a high-burden setting. The mixed methods approach and focus on real-world programmatic data make it valuable for TB/HIV program implementers. The study aligns well with the WHO and PEPFAR goals for TPT uptake and scale-up. Below are my comments:
- In the introduction: could the authors clarify what was the TPT coverage before the study?
- If possible could you provide a conceptual framework such as citing the implementation science models such as CFIR or health systems framework to enrich the interpretation of barriers and facilitators?
- Could the authors clarify how patient selection (which was as patient visited) reduced selection bias?
- Was the sample size adequate to power the study?
- Could you add a section or table clarifying the key TPT indicators?
- Please clarify the covariates included in the multivariable model and the rationale for their selection.
- what was the criteria for significance in this study analysis plan? (?p<0.05)
- How were missing data handled; as this is usually a significant issues in studies using programmatic data?
- What coding strategy was used in the qualitative analysis: deductive, inductive or hybrid? It should be clarified in the manuscript
- How was data saturation determined for this study? It should be clarified in the manuscript?
- Did patient speak in their local language, was there any translation that took place?
- How do the findings compare to regional or glabal evidence?
- Could the high completion rates be related to reporting bias?
- Could the author offer some specific, implementable strategies based on the study findings? What are the author's recommendations for future research?
- In the limitations could the authors talk about how missing data may have affected some analysis? Also mention potential recall bias during interviews.
Author Response
Comment 1: In the introduction: could the authors clarify what was the TPT coverage before the study?
Response 1: Thank you for the comment. There was no study specifically looking at the TPT coverage in these specific ART centres of Uganda. However, we have now mentioned the TPT coverage (6.9%) of Uganda as reported in the WHO global TB report.
Comment 2: If possible, could you provide a conceptual framework such as citing the implementation science models such as CFIR or health systems framework to enrich the interpretation of barriers and facilitators?
Response 2: Thank you for the comment. We used an explanatory approach to understand the facilitators and barriers for delivering TPT in the routine care setting. Also, the quantitative data was entirely based on the routinely collected programmatic data (secondary data). As the TPT for PLHIV was not newly introduced and implemented, application of implementation science framework like CFIR was not possible. Given this limitation, we had to adopt generic explanatory mixed-methods design and will not be able to apply any implementation science frameworks retrospectively.
Comment 3: Could the authors clarify how patient selection (which was as patient visited) reduced selection bias?
Response 3: Thank you for the comment. We really appreciate this comment, which has made us think about the potential limitation that we missed. Selecting only those who visited the health facilities during the recruitment period might have led of selection of PLHIV who might be relatively adherent to the health facility visits and might have had higher chance of initiating and completing their TPT course. Thus, the selection bias might have contributed to overestimation of TPT initiation and completion. We have now highlighted this as an additional limitation in the manuscript.
Comment 4: Was the sample size adequate to power the study?
Response 4: Thank you for the comment. The sample size was calculated for estimating the TPT initiation and completion rates. As we did not have a priori hypothesis for analytical inferential analysis (factors associated with initiation), a priori sample size calculation did not accommodate for power calculation. As exploratory model was attempted with limited number of independent variables, we found the sample size to be adequate for analysis with a thumb rule of 10 outcomes for each of the dummy variables included (16 categories for 519 PLHIV initiated on treatment).
Comment 5: Could you add a section or table clarifying the key TPT indicators?
Response 5: Thank you for the comment. The key TPT indicators for monitoring according the global TB report are TPT initiation rate and completion rate. All the other indicators (Percentage of PLHIV who don’t have TB at the start of ART, Percentage of PLHIV without active TB who underwent 4S screening at ART initiation, Percentage PLHIV with presumptive TB who underwent TB evaluation, Percentage eligible for TPT) are the process indicators that have an effect on the TPT initiation and completion rate. This has been now included in the methods section of the manuscript.
Comment 6: Please clarify the covariates included in the multivariable model and the rationale for their selection.
Response 6: Thank you for the comment. As this was an exploratory model, we included all the variables in the multivariate model. This has now been indicated in the data analysis section and also as a footnote in the table
Comment 7: What was the criteria for significance in this study analysis plan? (?p<0.05)
Response 7: Thank you for the comment. We used p<0.05 for data analysis. This has now been mentioned under data analysis.
Comment 8: How were missing data handled; as this is usually a significant issue in studies using programmatic data?
Response 8: Thank you for the comment. As this was an operational research, we wanted to reflect the ground realities in recording and reporting. There were no missing data in the demographic variables. The potential missing of documentation in TPT initiation and completion were considered as non-receipt of the service. However, as mentioned in the manuscript, data on TPT initiation was triangulated from multiple registers to avoid any underestimation of TPT coverage.
Comment 9: What coding strategy was used in the qualitative analysis: deductive, inductive or hybrid? It should be clarified in the manuscript
Response 9: An inductive coding approach was applied, allowing themes and categories to emerge directly from the data. This facilitated a grounded understanding of the facilitators and barriers to TPT initiation and completion as perceived by participants. This has been added to the manuscript.
Comment 10: How was data saturation determined for this study? It should be clarified in the manuscript?
Response 10: Yes determined after six interviews The number of participants interviewed was guided by the saturation of findings. This has been included in the manuscript.
Comment 11: Did patient speak in their local language, was there any translation that took place?
Response 11: The interview guide was prepared both in English and the local languages (runyoro and luganda). The interviews that were conducted in the local languages were translated to English at the time of taking notes since the interviewers were multi lingual. This has been included in the methods section in the manuscript
Comment 12: How do the findings compare to regional or global evidence?
Response 12: Thank you for this comment. The discussion section has been modified to include a comparison with regional and global evidence. Specifically, we highlight that while TPT initiation rates in this study (74–87%) are similar to those reported in other African and Asian settings (typically 60–80%), the TPT completion rates observed (≥98%) are notably higher than the global average, which often falls below 90%. We also note that the barriers identified—such as pill burden, documentation gaps, and health system constraints—mirror those described in other low- and middle-income countries.
Comment 13: Could the high completion rates be related to reporting bias?
Response 13: Thank you for this comment. The high completion rates may partly reflect reporting bias, as completion status was based on patient self-report and register entries without objective verification (such as pill counts). This limitation has been acknowledged in the Discussion.
Comment 14: Could the author offer some specific, implementable strategies based on the study findings? What are the author's recommendations for future research?
Response 14: Thank you for this comment. Some recommended strategies have been added to the manuscript.
Comment 15: In the limitations could the authors talk about how missing data may have affected some analysis? Also mention potential recall bias during interviews.
Response 15: Thank you for the comment. In the first limitation, we have explicitly mentioned the potential threat of missing data on the overall study findings. Based on the reviewer’s comments, we have even justified with an example. Changes made in the limitation section of the manuscript. We have also added additional limitation related to recall bias in the manuscript (fourth limitation)
